# Mitochondrial Dysfunction-Associated Mechanisms in the Development of Chronic Liver Diseases

**DOI:** 10.3390/biology12101311

**Published:** 2023-10-05

**Authors:** Madan Kumar Arumugam, Thiyagarajan Gopal, Rakhee Rathnam Kalari Kandy, Lokesh Kumar Boopathy, Sathish Kumar Perumal, Murali Ganesan, Karuna Rasineni, Terrence M. Donohue, Natalia A. Osna, Kusum K. Kharbanda

**Affiliations:** 1Research Service, Veterans Affairs Nebraska-Western Iowa Health Care System, Omaha, NE 68105, USA; madankumarbio@gmail.com (M.K.A.); sperumal@unmc.edu (S.K.P.); murali.ganesan@unmc.edu (M.G.); nosna@unmc.edu (N.A.O.); 2Department of Internal Medicine, University of Nebraska Medical Center, Omaha, NE 68198, USA; 3Cancer Biology Lab, Centre for Molecular and Nanomedical Sciences, Sathyabama Institute of Science and Technology, Chennai 600119, Tamil Nadu, India; 4Centre for Laboratory Animal Technology and Research, Sathyabama Institute of Science and Technology, Chennai 600119, Tamil Nadu, India; mailthiyagarajang@gmail.com (T.G.); lokeshkumarunom@gmail.com (L.K.B.); 5Department of Biochemistry and Molecular Biology, University of Maryland, Baltimore, MD 21201, USA; rakheerathnam@gmail.com; 6Department of Biochemistry and Molecular Biology, University of Nebraska Medical Center, Omaha, NE 68198, USA; karuna.rasineni@unmc.edu

**Keywords:** mitochondria, metabolic-dysfunction-associated steatotic liver disease, alcohol-associated liver disease, liver, hepatocytes, steatotic liver diseases

## Abstract

**Simple Summary:**

Mitochondria are crucially important organelles involved in various metabolic activities, including energy generation. The involvement of mitochondrial dysfunction in the etiology of major chronic liver diseases, including alcohol-associated liver disease and metabolic-dysfunction-associated steatotic liver disease, is receiving increasing attention. This review summarizes the current literature on common mitochondrial defects, including the enhanced production of mitochondrial reactive oxygen species, impaired ATP production and mitochondria-mediated inflammatory responses and cell injury/death. Understanding mitochondrial dysfunction and its involvement in the pathogeneses of chronic liver diseases is important for developing innovative and efficient treatment options.

**Abstract:**

The liver is a major metabolic organ that performs many essential biological functions such as detoxification and the synthesis of proteins and biochemicals necessary for digestion and growth. Any disruption in normal liver function can lead to the development of more severe liver disorders. Overall, about 3 million Americans have some type of liver disease and 5.5 million people have progressive liver disease or cirrhosis, in which scar tissue replaces the healthy liver tissue. An estimated 20% to 30% of adults have excess fat in their livers, a condition called steatosis. The most common etiologies for steatosis development are (1) high caloric intake that causes non-alcoholic fatty liver disease (NAFLD) and (2) excessive alcohol consumption, which results in alcohol-associated liver disease (ALD). NAFLD is now termed “metabolic-dysfunction-associated steatotic liver disease” (MASLD), which reflects its association with the metabolic syndrome and conditions including diabetes, high blood pressure, high cholesterol and obesity. ALD represents a spectrum of liver injury that ranges from hepatic steatosis to more advanced liver pathologies, including alcoholic hepatitis (AH), alcohol-associated cirrhosis (AC) and acute AH, presenting as acute-on-chronic liver failure. The predominant liver cells, hepatocytes, comprise more than 70% of the total liver mass in human adults and are the basic metabolic cells. Mitochondria are intracellular organelles that are the principal sources of energy in hepatocytes and play a major role in oxidative metabolism and sustaining liver cell energy needs. In addition to regulating cellular energy homeostasis, mitochondria perform other key physiologic and metabolic activities, including ion homeostasis, reactive oxygen species (ROS) generation, redox signaling and participation in cell injury/death. Here, we discuss the main mechanism of mitochondrial dysfunction in chronic liver disease and some treatment strategies available for targeting mitochondria.

## 1. Introduction

Mitochondria are the energy organelles of eukaryotic cells. They regulate cellular energy homeostasis and perform other key physiologic and metabolic activities including participation in cell death, ion homeostasis, ROS generation and redox signaling [1]. Key components of the mitochondrion are the inner mitochondrial membrane (IMM), the outer mitochondrial membrane (OMM), the intermembrane space and the mitochondrial matrix. The OMM contains porins that are necessary for material transport into and out of the organelle, whereas the highly-folded IMM forms the mitochondrial cristae, which harbor proteins of the electron transport chain (ETC). The latter proteins are involved in the oxidation of pyruvate and fatty acids presented to the mitochondrion [2]. Mitochondria supply about 90% of cellular energy in the form of adenosine triphosphate (ATP), which is crucial for cell growth and viability and is principally derived from the intracellular catabolism of fatty acids and carbohydrates [3]. Any dysregulation of mitochondrial function reduces the supply of cellular energy and leads to the accumulation of harmful levels of reactive oxygen species (ROS). Mitochondria also play important roles in maintaining redox balance, calcium homeostasis and in regulating their own turnover (by mitophagy, a lysosome-dependent method of mitochondrial destruction), thereby minimizing the numbers of dysfunctional mitochondria [4].

## 2. Factors Impairing Mitochondrial Function and Its Consequences 

### 2.1. Calcium Homeostasis under Physiological Conditions and Its Overload-Associated ROS Production and Cell Death

Calcium (Ca^2+^) is crucial for the optimal functions of many organelles, particularly the mitochondrion and the endoplasmic reticulum (ER). The movement of Ca^2+^ into and out of mitochondria is regulated by several pores and channels, including the voltage-dependent anion channel 1 (VDAC). The latter is located in the OMM and regulates Ca^2+^ flux into the mitochondrion. The interaction of the VDAC with the ER receptor that binds inositol 1,4,5-trisphosphate, controls the entry of Ca^2+^, pyruvate, reduced nicotinamide adenine dinucleotide (NADH), and ATP-into the mitochondrial inner-membrane space [5,6]. Another selective channel located in the IMM, the mitochondrial calcium uniporter (MCU), regulates the entry of Ca^2+^ into the mitochondrial matrix [7]. A similar but distinct form of MCU executes a mechanism called “rapid mitochondrial calcium uptake” (RaM). This allows for a rapid increase in Ca^2+^ uptake (within a few milliseconds) into the mitochondrial matrix. It is designed to rapidly induce ATP synthesis thereby rapidly responding to the hepatocyte’s energy needs [8]. MCU gene deletion investigations in mouse livers revealed that mitochondrial Ca^2+^ uptake is crucial for efficient hepatic lipid metabolism [9]. Notably, Ca^2+^ signaling and, particularly, the calcium levels in mitochondria, are crucial for liver regeneration [10].

However, Ca^2+^ overload (accumulation) negatively impacts mitochondrial function, leading to reduced ATP production and enhanced release of ROS [11]. Mitochondria appear to play a pivotal role in the regulation of two processes: one is apoptosis; the other is necrosis [12]. Apoptosis is the process of programmed cell death, whereby excessive Ca^2+^ release from the ER causes excessive Ca^2+^ uptake by mitochondria [13]. Accelerated reactive oxygen species (ROS) production by the mitochondrial respiratory chain ultimately depolarizes the inner mitochondrial membrane (IMM) mediated by the opening of the mitochondrial permeability transition pore (mPTP). Because of increased mitochondrial membrane permeability, a significant amount of mitochondrial Ca^2+^ and cytochrome c are released into the cytosol. Both these are important cell death inducers because they activate caspase-mediated apoptosis as shown in Figure 1. Interestingly, cyclophilin D, a mitochondrial matrix chaperone protein, regulates the opening of mPTP [14,15].

Necrosis is the cellular process of activating death receptors, including Fas and TNFα. Cellular stress triggers the activation of crucial molecules involved in the process of cell death. The receptor interacting protein (RIP) kinases, including RIP1 and RIP3, influence mitochondrial function by augmenting ROS generation via NADPH oxidase. Additionally, (ADP-ribose) polymerase-1 (PARP1) triggered by necrotic death stimuli, either activate RIP kinases or produce poly (ADP-ribose) (PAR) polymers. Similar to the process by which apoptosis is initiated, necrosis is also triggered by mitochondrial dysfunction through enhancing the uptake of Ca^2+^ and synthesis of ROS by the mitochondrial ETC, specifically complex I. This results in the activation and opening of mPTP [16,17]. A study using cyclophilin D-deficient mice clearly indicates that mPTP opening is central to the initiation of necrosis [17]. Paraptosis, a non-apoptotic form of cell death, is distinct from apoptosis in terms of morphology and biochemistry. This type of cell death occurs via cytoplasmic vacuolation with an enlargement of mitochondria or ER, rather than by pyknosis, DNA fragmentation or the activation of caspase [18,19,20]. The death of paraptotic hepatocytes is caused by the release of Ca^2+^ from the ER and its subsequent influx into mitochondria via the uniporter [19,21,22,23,24].

### 2.2. Dysregulated Iron Homeostasis 

Hepatic iron overload, which occurs in some metabolic-dysfunction-associated steatohepatitis (MASH) patients, influences the conversion of hydrogen peroxide (H_2_O_2_) to highly toxic hydroxyl radicals (HO•) via the Fenton reaction. Even mildly elevated levels of HO• within the cell will lead to cytotoxicity and oxidant stress [25]. Serum ferritin is an independent predictor of histologic severity and advanced fibrosis in patients with MASLD, and ferritin overload increases the risk of death [26,27]. Similarly, Buzzetti et al. found that hepatic iron deposition is associated with steatohepatitis, and serum ferritin increases with the progression of fibrosis to a pre-cirrhotic stage [28]. Recently, the association of serum ferritin level for assessing various stages of MASLD has been more clearly explained and is now considered a non-invasive biomarker for chronic liver diseases [28,29]. 

Using a rat model of steatohepatitis, investigators showed worsening steatosis and fibrosis after iron administration [30]. Furthermore, polymorphisms in the A736V TMPRSS6 gene variant influenced hepatic iron accumulation in patients with MASLD by regulating transcription of the mRNA encoding the hepatic hormone, hepcidin [31]. Cornejo et al. demonstrated that chronic iron overload in rats enhances inducible nitric oxide synthase (iNOS) in the liver by activating extracellular signal-regulated kinase (ERK1/2) and nuclear transcription factor kappa B (NFκB). Iron overload increases liver protein carbonylation by reducing the antioxidant capacity via depleting liver glutathione (GSH) content and lowering glutathione peroxidase (GPx) activity [32,33]. 

Recent evidence indicates that dysregulated iron homeostasis plays an important role in the progression of chronic liver disease via ferroptosis. Ferroptosis is an iron-dependent form of non-apoptotic cell death associated with excessive levels of lipid peroxides, and is distinct from other classes of cell death, including apoptosis and pyroptosis [34,35,36]. Besides the dysregulation of iron metabolism, other key cellular redox systems, including selenium-dependent GPx and the NAD(P)H/ferroptosis suppressor protein-1/ubiquinone axis, are involved in the ferroptosis-mediated progression of liver disease [37]. Yu et al. recently showed that hepatic *SLC39A14* deletion exacerbates iron-stimulated ferroptosis in mice with hepatic knockout of the transferrin *Trf* gene, which encodes the iron-binding protein, transferrin [38]. Studies on mouse models show that ferroptosis was found to initiate inflammation and modulate lipid peroxidation in the early stages of MASH [39,40]. Ferroptosis inhibitors were found to suppress the onset of diet-induced inflammation in MASH [41]. Ferroptosis inhibitors including ferrostatin-1, UAMC-3203 and VBIT-12, the VDAC1 oligomerization inhibitor, reportedly protect against acetaminophen-induced acute liver injury and ameliorate mitochondrial dysfunction [42,43]. The characterization of mitochondrial dysfunction and its contribution in the development of chronic liver diseases are listed (Table 1).

### 2.3. Excessive Carbohydrates and Fatty Acids Intake

Hepatic metabolic dysfunction is often associated with obesity, which is classically defined as the ratio of an excessive caloric intake and reduced energy expenditure or physical activity. Moreover, an excessive carbohydrate load increases hepatic de novo lipogenesis. The latter, combined with the increased hepatic uptake of adipose lipolysis-derived non-esterified free fatty acids (NEFAs), induces fatty acid overload in the liver. It is reported that about 80% of type 2 diabetes patients are obese and 45 to 75% have MASLD [43,44,45]. Despite these multiple metabolic insults, mitochondria acclimate by increasing their import and oxidation of NEFA by enhancing β-oxidation [46]. Interestingly, an inefficient hepatic mitochondrial β-oxidation capacity enhances fatty acid oxidation in microsomes and peroxisomes, causing oxidant stress and cellular damage [47]. Still, all these compensatory mechanisms are not enough to handle the increased hepatic NEFA overload. Therefore, fatty acids are condensed into triglycerides, contributing to liver steatosis, while mitochondria continue to acclimate by increasing their sizes, their rates of biogenesis and their oxidative capacities. Indeed, studies in rodents as well as in patients with early-stage steatotic liver diseases, have shown higher hepatic mitochondrial biogenesis and function in insulin-resistant obese subjects, compared with controls [48,49,50,51].

### 2.4. Electron Flux in the Electron Transport Chain, Electron Leakage and ROS Production

The mitochondrial electron transport chain is indispensable and fundamental to cellular metabolism. The system comprises five enzymatic complexes. The entire mechanism of ATP production is referred to as oxidative phosphorylation (OXPHOS). During OXPHOS, electrons generated from reducing substrates are passed to O_2_ through a chain of respiratory proton pumps. After glucose and fatty acids derived from dietary carbohydrates and lipids, respectively, are oxidized in mitochondria, the energy stored in these molecules is released as high-energy electrons via the TCA cycle and β-oxidation. These electrons are taken up by NAD and flavin adenine dinucleotide (FAD), leading to the production of NADH and FADH_2_, respectively. Then, NAD(H) and FADH_2_ molecules donate these high-energy electrons to the ETC [52]. The ETC is well characterized for its role in ATP generation and is exclusively present in the IMM. The ETC is comprised of four electron transfer chain complexes: I to IV: (CI-CIV) and ATP synthase (complex V) as the electron carrier [53]. Complex I, NADH-Q oxidoreductase, is an iron–sulfur-containing enzyme with a FMN (flavin mononucleotide) prosthetic group. This complex receives two electrons from the NADH. Complex II, succinate-Q reductase, accepts electrons from FADH2. Complexes I and II pass electrons to ubiquinone (Q), which is also known as coenzyme Q (CoQ). Complex III, cytochrome c reductase, is composed of the iron–sulfur (Fe-S) proteins, cytochrome b and cytochrome c. This complex accepts electrons from the CoQ and directly passes these to complex IV, which is cytochrome c oxidase. Electrons are finally transferred onto O_2_ by complex IV to form water (H_2_O). The reduction of one molecule of oxygen (O_2_) requires four electrons. As electrons pass freely through these transporter chains, the free energy released is used to pump protons (H^+^) from the mitochondrial matrix to the intermembrane space, creating a proton gradient. Protons diffuse along this electrochemical gradient at complex V, which has two heme groups (heme a and a3) and two copper (Cu^2+^A and Cu^2+^B) catalytic centers. Energy released by the proton and about 90% of molecular O_2_ are utilized by this complex to generate ATP from ADP [53]. On the other hand, uncoupled mitochondrial respiration due to H^+^ leak is mediated by adenine nucleotide translocase (ANT) and uncoupling proteins (UCP), which dissipate membrane potential with the loss of ATP production [54]. Some electrons are directly transferred to O_2_ ROS in the ETC. Thus, mitochondria play a critical role in energy metabolism, the failure of which leads to chronic disorders of energy metabolism, including fatty liver disease.

#### 2.4.1. Electron Leakage and ROS Production

The mitochondrial ETC involves mechanisms for electrons movement from the oxidation of substrates and ATP production. A small percentage of electrons in the ETC (~0.2–2%) do not follow the expected transfer to the final electron acceptor (O_2_) and are not completely associated with ATP production. Instead, their “leakage out” of the ETC generates superoxide (O^2•−^) or H_2_O_2_ after contact with oxygen [55,56]. With a total of 11 sites producing O^2•−^ and/or H_2_O_2_, the ETC found in mitochondria is closely tied to substrate oxidation (Figure 2) [57]. Increased ROS formation is associated with several chronic diseases, particularly in hypoxic conditions. Specifically, complex I (CI) and complex III (CIII) are thought to be primary sources of ROS generation in mitochondria [57,58]. According to Hernansanz-Agustin et al., cells that are exposed to CI in acute hypoxia produce powerful superoxide within minutes [59]. Although site CIIF produces very little ROS under normal circumstances, site CIIF is primarily responsible for the increases in ROS seen in CII mutation-related illnesses [60]. The availability of reduced flavoprotein as FAD, which is notorious for generating electron leakage and producing ROS in the mitochondrial matrix, is primarily responsible for the efficiency of site CIIF for ROS generation. [61]. Moreover, the influence of site CIIF is diminished by TCA cycle intermediates, especially oxaloacetate, malate and succinate, and thus attenuates to access O_2_ at site CIIF, leading to ROS production [62]. In comparison with ROS generation by CI, site CIII modest production of ROS could be ignored [57]. CIII uses the Q-cycle to transfer electrons, whereas a single electron carrying ubiquinone (QH), flowing freely within CIII directly leaks the electron to O_2_ by producing ROS non-enzymatically [63]. As a result, both the mitochondrial matrix and the intermembrane gap will receive the released ROS. CIV has been discovered to be less likely than the other complexes to generate ROS. [54].

#### 2.4.2. Blockade of ETC Complexes Suppresses ROS Generation

Rotenone and piericidin are well-known blockers of the quinone-binding site of complex I (IQ) and each diverts the transfer of electrons to ubiquinone (CoQ), thereby increasing ROS production at the flavin site of complex I (CIF). Quinlan et al. explained the mechanism of inhibition of the Q cycle by antimycin, resulting in the obstruction of electron flow to the quinone-oxidizing site of complex III (CIIIQo) and the production of ROS after reacting with O_2_ [64]. Likewise, stigmatellin and myxothiazol, quinone-oxidizing (Qo) site-specific antagonists, inhibit ubiquinol (QH_2_) binding to the Qo site, thereby preventing the production of ROS in CIII [64]. S3QELs, which are small molecule suppressors of site CIIIQo electron leak, selectively attenuate the production of superoxide and H_2_O_2_ radicals at CIII of the ETC without inhibiting oxidative phosphorylation. S3QELs, a chemical CIIIQo electron leak suppressor, was previously shown to inhibit the production of ROS at the CIIIQo site without impairing electron transport or the redox states of other centers [65].

### 2.5. Failure of Liver ROS Clearance Capacity

While some recent studies have clearly confirmed that ROS are likely crucial second messengers for several intracellular pathways [66,67], enhanced ROS generation has profound detrimental consequences. As mentioned earlier, the chief route of ROS generation by the ETC is the premature leak of electrons from complexes I, II and III to the one-electron reduction of molecular oxygen to O^2•−^. Several factors can significantly modulate O^2•−^ release from the ETC, as recently reviewed [54]. A rise in the intramitochondrial concentrations of H_2_O_2_ depends on two antioxidant systems in the mitochondrial matrix, reduced GSH and the thioredoxin (Trx)/peroxiredoxin (PRx) systems. The matrix localized Mn^2+^ superoxide dismutase (SOD) and the intermembrane space present Cu/Zn SOD dismutates the ETC generated superoxide to H_2_O_2_. Catalase, which inhabits liver peroxisomes then coverts H_2_O_2_ to H_2_O and O_2._ In addition, H_2_O_2_ and lipid oxy (LO•) or peroxyl radicals (LOO•) are neutralized by GPx/GSH, which acts as an electron donor. Similarly, H_2_O_2_ and LOO• are reduced by Prx using Trx as the electron donor [68,69]. The Prx/TrxR system operates in both the cytosol and the mitochondrial matrix [68]. NADPH are required to replenish their antioxidant/reductive properties. Furthermore, both glutathione reductase and thioredoxin reductase require NADPH for their reductive activities [70]. The level of reduced NADPH is balanced through catalysis by malic enzyme, glutamate dehydrogenase and isocitrate dehydrogenase, which are all located in the mitochondrial matrix [70]. 

#### 2.5.1. Deregulation of ROS Homeostasis Causes Progressive Liver Injury

Any dysfunction in the ETC produces excessive electron leak and increases ROS generation and eventual liver injury. An overproduction of ROS or failure to clear ROS can lead to the functional impairment of enzyme activities, and altered cellular functions, eventually resulting in oxidant stress [71]. Oxidant stress is the principal mechanism behind hepatocellular injury, which then triggers hepatic inflammation and fibrosis in MASLD and ALD [72,73,74]. Increased levels of ROS, which evoke the production of proinflammatory mediators, activates NFκB and nucleotide-binding oligomerization domain-like receptor family pyrin domain-containing 3 (NLRP3). These inflammasomes trigger the production of inflammatory cytokines, including IL-1β, IL-6 and TNF-α [75]. The latter event results in the recruitment of and infiltration by immune cells, including peripheral blood monocytes into the hepatic parenchyma, causing steatohepatitis [76,77]. Mitochondrial membrane damage by ROS causes the formation of mPTP and the release of mitochondrial DNA (mtDNA), both of which are called danger-associated molecular pattern(s) (DAMPs), which activate the NLRP3 inflammasome [78]. Moreover, proinflammatory cytokines stimulate hepatocyte apoptosis by activating initiators of death receptors including Fas and TNF-related apoptosis-inducing ligand (TRAIL) [79]. Recent findings indicate that genetic or epigenetic alterations, as well as impairments in metabolic and immunologic pathways, have been closely associated with inflammation, fibrosis and their subsequent development to hepatocellular carcinoma [79].

#### 2.5.2. Imbalance of ROS Homeostasis Aggravates Inflammation in ALD

Alcohol-associated aberrant mitochondrial ROS production contributes significantly to hepatocyte injury. Ethanol is oxidized to acetaldehyde by the cytosolic alcohol dehydrogenases (ADHs) and by the microsomal cytochrome P450 2E1 (CYP2E1)-mediated catalysis. CYP2E1 is an inducible enzyme, as its intracellular levels rise after continuous alcohol consumption because ethanol oxidation slows CYP2E1’s rate of degradation by the ubiquitin-proteasome system (UPS) [80]. Because of its rather unique catalytic cycle, CYP2E1 induction by ethanol increases the generation of ROS, (including hydroxyethyl and hydroxyl radicals), thereby enhancing cellular oxidant stress [81]. Moreover, the altered redox status triggered by augmentation of the NADH/NAD^+^ ratio enhances the formation of ferrous iron (Fe^2+^) from ferric iron (Fe^3+^). As discussed earlier, iron overload occurs in mitochondria, resulting in disrupted mitochondrial respiration. Therefore, increased liver ROS due to ethanol metabolism, damages mtDNA and disrupts the ETC, resulting in the release of mtDNA [82,83]. However, long-term alcohol consumption damages mitochondria and disrupts their biogenesis, resulting in lower numbers of functional mitochondria in hepatocytes [83,84]. In addition, acetaldehyde, the primary product of ethanol oxidation, has pro-inflammatory and fibrogenic properties and is a potent activator of steatohepatitis and fibrosis in ALD [85]. Furthermore, the ethanol-induced proinflammatory cytokines, mentioned earlier, are also involved in the activation of oncogenic signaling pathways, including signal transducer and activator of transcription 3 (STAT3)/Janus kinase 2 (JAK2) and c-Jun-N-terminal kinase (JNK). The latter activations can lead to malignant transition from MASH to HCC [79,86]. Taken together it implies that when the mitochondrial antioxidant system fails to balance surplus ROS generation, the increased ROS induce mitochondrial DAMP accelerates liver injury progression.

### 2.6. Mechanisms of Inefficient Respiratory Chain

It is well known that mitochondrial OXPHOS is a principal source of hepatocellular ATP synthesis in an aerobic environment. To maintain this energy homeostasis in the liver, mitochondria must be able to sense and respond promptly to alterations in nutritional status and energy requirements. The degree of coupling between substrate oxidation and phosphorylation is resilient and keenly regulated by the mitochondrion. When a cell is in a state of homeostasis, the efficiency of OXPHOS is importantly fixed so that the coupling of mitochondrial respiration (oxidation) and ATP production should occur to adequately fulfill the demand for cellular ATP. Conversely, when the coupling efficiency of mitochondrial respiration (substrate oxidation) varies, it cannot efficiently produce ATP in stoichiometric amounts. This latter is called “uncoupled” mitochondrial respiration (OXPHOS) [87]. Recent studies by Eyenga et al. showed that there is a direct association between ATP production and mitochondrial respiration that shifts during sepsis [88,89].

The mechanisms involved in the loss of coupling efficiency during mitochondrial respiration are: (1) proton leak through the IMM from the positive compartment to the negative (uncoupled proton leak); (2) activation of mPTP by cycling of other cations such as Ca^2+^; (3) alterations in redox proton pumping stoichiometry (CI, CIII and CIV) by redox and proton slipping, where H^+^ pumps express a variation in the quantity of H^+^ accumulated in the intermembrane space (redox slipping) and/or the amount of H^+^ return back to the matrix through ATP production by F1Fo ATPase (proton slipping); (4) decoupling of electron flow, which causes a loss of integrity of the cellular compartment; and (5) univalent reduction of molecular oxygen to superoxide anion radical synthesis during electron leak [87,89].

### 2.7. Hepatic Tricarboxylic Acid (TCA) Cycle and Mitochondrial Respiratory Efficiency

The tricarboxylic acid (TCA) cycle is also known as the citric acid cycle or the Krebs cycle. During mitochondrial respiration, the flow of electrons is coupled to phosphorylation to form ATP and the energy conversion is facilitated by the proton motive force. Electron transfer chains are fueled by in and out diffusion and by the movement of substrates through the IMM and OMM. In mitochondria, acetyl-CoA, generated from pyruvate during glycolysis (or from beta-oxidation of fatty acids), serves as a point of entry to generate citrate. Aconitase converts citrate to isocitrate, which, is converted to α-ketoglutarate by oxidative decarboxylation, producing CO_2_ as a byproduct, with the simultaneous reduction of NAD^+^ to NADH. Alpha-ketoglutarate is also produced from the deamination of glutamine to form glutamate, catalyzed by glutaminases I and II. Alpha-ketoglutarate is subsequently oxidized to succinyl-CoA, producing NADH and CO_2_. Next, succinyl-CoA is hydrolyzed to succinate, coupling it to produce energy as GTP, which can be converted to ATP. Succinate is oxidized to generate fumarate by succinate dehydrogenase/complex II, generating two FADH (or FADH_2_) by transferring two hydrogen atoms to FAD. Further, fumarate is hydrated to form malate by fumarase and then malate is oxidized to oxaloacetate and NADH. Finally, oxaloacetate undergoes an aldol condensation to yield acetyl-CoA and H_2_O, thereby continuing the TCA cycle. Overall, in the TCA cycle, the reduction of NAD^+^ to NADH is coupled to produce CO_2_. In contrast, β-oxidation of fatty acids generates reducing equivalents through the following reactions: (1) FADH_2_ serves as the substrate of electron-transferring flavoprotein complex, (2) synthesis of acetyl-CoA by chain shortening, and (3) production of NADH, catalyzed by 3-hydroxyacyl-CoA dehydrogenases. ATP generation is dependent on acetyl-CoA, which enters the TCA cycle or is used to generate ketone bodies via ketogenesis [90].

Several investigations report that mitochondria from livers of animals as well as patients with MASLD have enhanced TCA cycle function, with diminished respiratory coupling in their liver mitochondria [91,92]. Anaplerotic/cataplerotic fluxes, the nonoxidative fluxes of intermediary metabolites into and out the TCA cycle, are reportedly augmented in MASLD [93]. When this increased mitochondrial metabolic flux is associated with inefficient oxidative capacity, it triggers production of higher ROS levels with subsequent hepatic inflammation, thereby contributing to hepatocellular injury [93,94]. An augmented fatty acid flux into hepatocytes during insulin resistance was shown to amplify the TCA cycle and anaplerosis [93]. Although obesogenic subjects had higher numbers of mitochondria with elevated oxygen consumption, only patients with a diagnosis of MASH exhibited mitochondrial coupling and leaking activity with elevated hepatic oxidant stress, ROS production and hepatic injury [95]. Similarly, mice fed a high-fat diet showed augmented TCA cycle activity [94].

The condensation of acetyl-CoA from β-oxidation into ketogenesis results in an impairment in the mitochondrial matrix which significantly impacts hepatic lipid and glucose metabolism to promote MASLD pathogenesis [96]. Deregulated ketogenesis in rodents promotes fat accumulation and induces the TCA cycle with upregulated gluconeogenesis and *de novo* lipogenesis, leading to pathological features of chronic liver injury [97,98]. Fletcher et al. reported that ketogenesis, particularly the generation of β-hydroxybutyrate, worsened after triglycerides accumulated in liver. MASLD patients with resistance to ketosis exhibit no decline in their levels of β-oxidation, whereas a positive correlation was observed with elevated acetyl-CoA oxidation in the TCA cycle, higher mitochondrial respiration and an increased rate of gluconeogenesis. These findings clearly indicate that in MASLD, hepatic tissue accelerates oxidative metabolism instead of selectively clearing the surplus amount of NEFA as ketone bodies [99]. 

A 50% increase in hepatic mitochondrial anaplerosis was reported in humans with high intrahepatic triglycerides compared with those patients with low hepatic lipids, signifying that elevated TCA cycle flux occurred via increased levels of enzymes involved in gluconeogenesis, including pyruvate carboxylase and phosphoenolpyruvate carboxykinase in MASLD subjects [100]. Dysregulated insulin signaling was also observed by augmented gluconeogenesis and anaplerosis with increased oxidative TCA cycle flux. Therefore, the TCA cycle seems to allow for hepatic mitochondrial malfunction during insulin resistance by augmenting electron accumulation into an inefficient respiratory chain, thereby forming greater levels of ROS and production of mitochondria-mediated substrates for increased gluconeogenesis [91]. One study demonstrated that treatment with pioglitazone, an insulin sensitizer and PPARγ-agonist, reduced TCA cycle flux and anaplerosis in rodent models of MASH, suggesting that pioglitazone could be used therapeutically to ameliorate hepatic mitochondrial oxidative dysfunction and steatohepatitis development [101].

### 2.8. Hepatic Inflammation

Inflammation is a defense mechanism that the body deploys to maintain homeostasis, in response to invading pathogens, ischemic and physical injury, exposure to toxins and irradiation or different types of traumas [102]. However, inflammation is a double-edged sword, with both beneficial and pathological consequences. Deregulated or unresolved inflammation acts as a ***disease driver*** that results in various human pathologies associated with the nervous, cardiovascular, hepatic and renal systems. In addition, uncontrolled chronic inflammatory responses can ultimately result in neoplasia, thereby expediting tumor progression by suppressing cancer immunosurveillance [103,104,105,106,107]. The involvement of the immune system is critical to the process of inflammation. It usually follows a series of reactions that occur after the initiation and activation of pattern recognition receptors (PRRs) expressed by immune and non-immune cells of the host, which detect distinct evolutionary conserved structures on pathogens or DAMPs released from damaged/dying host cells [108,109]. Toll-like receptors (TLRs), NOD-like receptors (NLRs), C-type lectin receptors (CLRs), and retinoic acid inducible gene-I (RIG-I)-like receptors (RLRs) are the four distinct PRR classes [110]. Under certain circumstances, the endogenous molecules, including, nucleic acids, proteins like calreticulin and other metabolites like ATP, fail to drive PRR-mediated signaling due the restricted access to the PRR subcellular compartments [111]. 

#### 2.8.1. Role of Mitochondria in the Inflammatory Response

Decades of research have documented that, in addition to their classical role in energy production and metabolism, mitochondria participate in a diverse range of biological activities, including inflammation. The mitochondrial genome, potentially stemming from its bacterial origin, is a potent stimulator of innate immunity and it plays a significant role in driving inflammation [112]. MtDNA is especially vulnerable to oxidation, as it is in close physical proximity to a major source of ROS, the ETC, potentially resulting in mtDNA mutations. Furthermore, unlike nuclear DNA, mtDNA is hypomethylated and may be sensed as foreign. The presence of oxidatively damaged and unmethylated CpG motifs in endogenous mtDNA upon release into the cytoplasm and out into the extracellular milieu activates PRRs to elicit various inflammatory pathways, causing a plethora of detrimental biological consequences, including infection and cell death [113,114]. The integrated signaling networks operated by mtDNA in the inflammatory process have been targeted in the clinical management of various human pathologies [115]. The presence of pathogen-derived cellular and mtDNA in the cytosol triggers the activation of various immunogenic pathways that signals via cyclic GMP-AMP (cGAMP) synthetase (cGAS), TLR9 and NLRP3 inflammasome [116].

The cGAS–stimulator of interferon genes (STING) pathway activates innate and adaptive immunity through type I interferon response. cGAS is one of the cytosolic DNA sensors that produces cGAMP, a second messenger derived from ATP and GTP, in response to DNA binding. Further, the binding of cGAMP to the endoplasmic reticulum membrane adaptor STING results in conformational changes and the subsequent activation of STING. Activated STING then translocates to the Golgi compartments and an ER-Golgi intermediate apparatus from the ER. Simultaneously, the transcription factor interferon regulatory factor 3 (IRF3) is phosphorylated by TANK-binding kinase-1 (TBK1) recruited through the carboxy terminal PLPLRT/SD motif of STING, ultimately resulting in a type I interferon response and the further activation of the innate and adaptive immune response through the dimerization and translocation of phosphorylated IRF3 into the nucleus [117,118,119,120,121,122].

Activated type I interferon responses have been documented during mitochondrial apoptosis upon caspase inhibition. Following the mitochondrial outer membrane permeabilization (MOPM), Rongvaux et al. observed type I interferon expression and interferon-stimulated gene response upregulation in mouse models and embryonic fibroblasts that lack caspases 3 and 7 or 9 [123]. Another investigation carried out by a different team revealed that deletion of caspase 9 in hematopoietic stem cells increased basal levels of type I interferons, and that inducing apoptosis in the same circumstances had the same consequences. [124]. Both studies corroborated that the increased interferon responses observed in their studies were operated through the mtDNA-mediated activation of the cGAS-STING pathway [123,124].

#### 2.8.2. Activation of TLR9

TLR brings about innate immunity by recognizing a myriad of heterogenous bacterial signatures. Immune cells, including macrophages, dendritic cells, monocytes and B-cells express TLR9. Primarily, the presence of TLR has been reported on the endoplasmic potent activator of TLR9 [125,126]. A study conducted revealed the presence of formyl peptide, a mitochondrial-derived PAMP and mtDNA in the plasma samples of trauma patients and individuals with injuries unrelated to any type of pathological infections. This concept was first reported by Zang et al., where they noted that the mtDNA released into the blood during the systemic inflammatory response syndrome could activate the TLR9 present on neutrophils, causing Ca^2+^ flux and p38 MAPK activation, eventually resulting in inflammation [127,128]. The binding of DNA to TLR9 results in the recruitment of an adaptor molecule, MyD88, and the further activation of mitogen-activated protein kinases (MAPK) and NFkB. The latter leads to the production of pro-inflammatory cytokines and chemokines, ultimately triggering inflammatory responses [129]. Moreover, the TLR9-MyD88 complex activates interferon regulatory factor 7 (IRF7), a transcription factor. This factor enhances production of type I interferon [130,131]. Several recent studies demonstrating inflammation induced by mtDNA in various human pathologies, including MASH, emphasize the role of mitochondrial dysfunction in mediating the inflammatory response [132,133,134,135,136].

#### 2.8.3. Activation of Inflammasome by Mitochondria

Inflammasomes are a group of cytosolic protein aggregates. They have been identified as an indispensable part of the host innate immune system, which consists of NLRP3, NOD-like receptor CARD domain-containing 4 (NLRC4) and absence in melanoma 2 (AIM2) (innate immune sensor molecule), as well as procaspase-1 (an effector molecule). Additional adaptor proteins like apoptosis-like speck protein or ASC may be required to bridge the protein–protein (CARD-CARD) interactions and enhance the complete assembly of inflammasomes under certain circumstances. Following an infection or injury, the activators are stimulated, which, in turn, leads to the formation of a large cytosolic protein complex consisting of an inflammasome sensor, adaptor and effector, which ultimately causes the self-cleavage and activation of procaspase 1. The activated caspase, thus produced, processes the immature pro-IL-1β and pro-IL-18 to their biologically active forms, thereby activating the inflammatory response [137,138,139,140,141].

The mtDNA-controlled activation of the NLRP3 inflammasome has been well-documented by studies conducted in animal models [142]. Additionally, the selective deletion of autophagy-related proteins, beclin-1 (an upstream autophagy regulator) and microtubule-associated protein 1 light chain 3B (LC3B), followed by subsequent treatment with exogenous lipopolysaccharide (LPS) and ATP, enhances the accumulation of damaged, ROS-generating mitochondria, cytoplasmic translocation of mtDNA and activation of NLRP3-mediated mechanisms, resulting in the production of IL-1β and IL-18 [143]. In line with this, bacterial-sepsis-related mortality and elevated levels of IL-1β in the circulation have been reported in mice deficient in autophagy gene, *Map1lc3b*, which encodes microtubule-associated proteins 1A/1B light chain 3B, compared with wild type animals. Further, an earlier study conducted by Saitoh et al. demonstrated that a deficiency in autophagy-related 16-like 1 (Atg16L1) protein in macrophages was associated with enhanced endotoxin-induced production of IL-1β and IL-18 [144]. 

#### 2.8.4. Role of Mitochondria in the Activation and Release of Pro-Inflammatory Cytokines

NLRP3 activation involves two specific steps that include priming and activation. The engagement of TLRs by the PAMPs and subsequent NFkB activation-mediated *de novo* synthesis of pro IL-1β and NLRP3 expression describe the priming step. Zhong et al. reported that priming induces DNA polymerase γ-catalyzed synthesis of new mtDNA [145], whereas the activation step involves caspase-1-dependent proteolytic processing- mediated NLRP3 inflammasome assembly. The engagement of TLR signals via MyD88/TRIF leads to activation of the transcription factor interferon regulatory factor 1 (IRF1), followed by induction of the mitochondrial nucleotide kinase cytidine/uridine monophosphate kinase 2 (CMPK2) expression for the synthesis of mtDNA. Not packaged by mitochondrial transcription factor A (TFAM), the newly synthesized mtDNA is more susceptible to nuclease action and oxidative damage and is a major source of oxidized mtDNA and NLRP3 inflammasomes activation. The activation of the parkin-p62-dependent mitophagy pathway restricts NLRP3 inflammasome activation by eliminating damaged mitochondria to orchestrate a self-limiting host response and maintain homeostasis [146].

#### 2.8.5. Mitochondrial-Targeted Treatments for Inflammatory Responses

The explosion of knowledge in past decades demonstrates the involvement of mitochondria in inflammation-associated human pathologies. This has led to the development of therapies that target the effector phases of inflammation such as PRR, the use of certain agonists of the signal transducers (e.g., STING1 agonist) or the use of neutralizing antibodies/cytokines for clinical management of the pathology [115,147,148,149,150].

Other therapeutic strategies include targeting specific mitochondrial functions, including MOMP and mitochondrial permeability transition, that may benefit patients with a better outcome. Venetoclax, a selective inhibitor of BCL-2, is the only licensed drug to be used in humans that target the molecular machinery of MOMP. Currently, the inhibitor is used in treating a few hematological malignancies, and its efficacy is being tested for solid tumors [115]. In addition, other BCL-2 inhibitors (navitoclax and ABT-737) were developed, but thrombocytopenia development in treated patients has caused ABT-737 to be no longer prescribed [151,152]. Approaches that utilize BAX activators to initiate MOMP to treat apoptosis-resistant cancers are also available [153,154,155,156]. Importantly, cyclosporine A, an approved immunosuppressive drug for humans, exerts its effect by blocking mitochondrial permeability transition and subsequent inflammatory reactions (Figure 3) [149].

#### 2.8.6. Role of PGC-1α on Inflammatory Response

The peroxisome proliferator-activated receptor gamma coactivator-1 alpha (PGC-1α) regulates the expression of genes involved in fatty acid oxidation and OXPHOS [157,158]. PGC-1α has a protective impact against inflammation in liver tissue [159]. It controls the metabolic circuit that connects oxidative stress, inflammatory response and metabolic syndrome and is a modulator of cellular respiration, uptake of energy substrates and mitochondrial biogenesis [157,160]. It accomplishes these multifaceted effects by linking itself to a variety of proteins that regulate cellular metabolism, such as peroxisome proliferator-activated receptors (PPARs), nuclear respiratory factor–1/2, farnesoid X receptor, liver X receptor, forkhead box protein O1 (FOXO1), sirtuin 1 (Sirt1), tumor protein p53 and hepatocyte nuclear factor 4. Wang et al. showed that increasing PGC-1α mRNA expression stimulates mitochondrial activity and ameliorates liver injury by increasing the IL-10-mediated anti-inflammatory response [161]. Further studies revealed that PGC-1α-mediated mitochondrial biogenesis plays a vital role in mitigating the high-fat diet-induced hepatic steatosis development [162]. Wan et al. corroborated these findings and further clarified that the PGC-1α protection against the development of hepatic steatosis and insulin resistance is by enhancing IL-10-mediated anti-inflammatory responses [163]. Indeed, suppressing PGC-1α expression increased IκBα phosphorylation, which in turn increased NFκB p65 nuclear translocation as shown in Figure 4 [163,164]. Collectively, these findings imply that PGC-1α may control lipid metabolism by limiting inflammation and by stabilizing the complex IκBα/NFκB in the cytoplasm. Further supporting evidence is provided by many studies demonstrating that PGC-1α reduces the expression of proinflammatory cytokines induced by various inflammatory stimuli, including TNF, TLR agonists, and saturated FFAs [164,165,166,167].

### 2.9. Mitochondrial Protein Methylation—Fusion and Fission

Post-translational methylation is a common protein modification process that is catalyzed by a variety of methyltransferases (MTases) [168]. Protein methylation is crucial for optimizing and controlling cellular and physiological functions, as it alters protein activity. Research has generally focused on the methylation of nuclear and cytoplasmic proteins and their role in health and disease. Lysine and arginine residues are the most typical targets of protein methylation, although other residues, including glutamine and histidine, can also undergo methylation [72]. These residues become more substantial and have altered hydrogen-bonding abilities after they are methylated, but their charges are unaffected. Not much is known regarding mitochondrial proteins, which are also methylated. However, in the last decade, there has been significant progress in understanding the functional significance of mitochondrial protein methylation and the MTases that catalyze this reaction. This had led to the discovery of many unique mammalian MTases that methylate lysine, arginine, histidine and glutamine residues in diverse mitochondrial protein substrates [169,170]. A few of these proteins, including citrate synthase, ATP synthase and respiratory complex I, are important parts of the bioenergetic machinery [171]. Since some or all of the citric acid cycle enzymes, including citrate synthase, are arranged in metabolons within mitochondria for ease of substrate flow among the involved enzymes, alterations in methylation status (such as n Lys-368 on citrate synthase), by affecting metabolon’s protein–protein interactions or the regulation of substrate channeling [171,172], could impair not only energy production, but also cell metabolism and fate. Defective fatty acid β-oxidation in liver mitochondria is a key process which is impaired in both ALD and MASLD [72,73,173]. Furthermore, an alcohol or high-caloric intake-induced increase in circulating free fatty acids and the lipotoxicity associated with liver uptake as well as the imbalance between mitochondrial ROS generation and impaired antioxidant content/activity (SOD, GPx, mitochondrial GSH) are additional mechanisms that contribute to mitochondrial dysfunction during ALD and MASLD pathogenesis [72,73,173,174,175,176].

Mitochondrial fusion and fission help to maintain organelle homeostasis and allow mitochondrial metabolism to be precisely tuned to cellular needs [177]. The two most frequent alterations in mitochondrial morphology are fission, the generation of mitochondria from a larger mitochondrion, and the fusion of two different mitochondria. “Mitochondrial dynamics” refers to synchronized cycles of fusion and fission that regulate various aspects of mitochondrial function and support cellular homeostasis. A unique relationship exists between mitochondrial dynamics and metabolism, and each impacts the other. All cells undergo fusion and fission events, albeit the frequency varies depending on the type of cell. These activities are crucial for maintaining adequate numbers of functional mitochondria. It is recognized that these alterations in mitochondrial morphology have an impact, especially on cellular function in hepatocytes, which contain high numbers of mitochondria. Mitochondrial fission and fusion play significant roles in mitochondrial homeostasis, mtDNA inheritance and intracellular distribution as those organelles undergoing fusion and fission are continuously adapting to changing physiological needs. In order to preserve cellular homeostasis, individual mitochondria or their components, such as OXPHOS protein complexes and lipids, are destroyed by mitophagy. Chronic oxidative shocks that overpower mitochondrial quality control (MQC) systems cause severe mitochondrial damage that is harmful to the survival and function of hepatocytes. Preclinical data suggest that MQC regulation is therapeutically advantageous against MASLD/MASH by the tight regulation of mitochondrial fission and fusion. Increased levels of ROS are accompanied by increased mitochondrial fission under circumstances of higher metabolic flux, leading to oxidative stress and cell death [177]. Notably, elevated ROS and cell death were prevented by suppressing mitochondrial fission in cells subjected to high levels of glucose or fatty acids. Therefore, it is likely that mitochondrial fission precedes increased ROS in metabolic overflow and that fission may be a target to ameliorate oxidant stress related to metabolic disorder. There is evidence of increased mitochondrial fission in fibrotic livers of humans and mice [178]. Overexpression of mitochondrial fisson1 protein (fis1, a component of a mitochondrial complex that promotes mitochondrial fission) activates hepatic stellate cells (HSCs). Indeed, increased mitochondrial fission is necessary for hepatic stellate cell (HSC) activation as evidenced by increased apoptosis of activated HSCs after blocking mitochondrial fission with the mitochondrial fission inhibitor-1, Mdivi-1. In addition, excessive mitochondrial fission and the downregulation of mitofusin-1 (Mfn1) reportedly promotes metastases of hepatocellular carcinomas [179]. In contrast, overexpressing Mfn1, which restores mitochondrial fusion, slows the growth and spread of hepatocellular cancer. The stability of mtDNA in hepatocytes has also been shown to depend on mitochondrial fusion. The proteins that regulate outer and inner mitochondrial membrane fusion play a major role in maintaining the stoichiometry of the protein components of the mtDNA replisome. However, it was discovered that chronic ethanol exposure completely eliminates the dynamics of mitochondrial fusion and motility in primary hepatocytes. Leptin, a hormone that regulates energy, can also affect the dynamics of mitochondria in hepatocytes by activating Mfn1 to enhance mitochondrial fusion. The latter reduces the buildup of fatty acids caused by high glucose in primary hepatocytes. Patients with extrahepatic cholestasis have lower endogenous levels of mitochondrial fusion protein-2 (Mfn2) [178]. The significance of changes in mitochondrial fusion in cholestatic liver injury is underscored by a significant reduction in mitochondrial fragmentation and repair of mitochondrial damage by overexpression of Mfn2. Changes in mitochondrial morphology, particularly mitochondrial fission, are typical hallmarks of liver injury of diverse etiologies. Hepatocytes most likely fuse and split mitochondria at a slower rate than cells from other tissues [180]. While the stimulation of mitochondrial fission may help mitophagy as an adaptive response to deal with early stress, further research is needed to fully understand the role of the autophagy/mitophagy, which may vary depending on the type of liver injury.

### 2.10. The Insufficiency of Antioxidants and Impaired ROS Clearance

Since increased oxidative-stress-mediated inflammation contributes to liver damage, antioxidants are regarded as an effective therapeutic approach for treating liver disease [24]. Elevated mitochondrial ROS levels are accompanied by inflammatory cytokines that impair cellular integrity [75]. Additionally, ROS are associated with a higher production of reactive aldehydes, which inhibit the mitochondrial respiratory chain and impair electron flow. As a result, there is increased mitochondrial production of ROS and oxidative stress [181,182]. Pro-oxidant overproduction results in harmful cellular processes, including lipid peroxidation, oxidative DNA damage and protein degradation, which triggers an increased production of cytokines, including TNF, IL-6 and transforming growth factor, resulting in fibrogenesis [183]. Enzymatic and non-enzymatic antioxidants are released by cells in response to oxidative stress under physiological circumstances. The majority of the body’s endogenous antioxidant system comprise of enzymatic antioxidants like SOD, catalase, and glutathione peroxidase, hydrophilic antioxidants like urate, ascorbate, GSH, and flavonoids, and lipophilic radical antioxidants like tocopherol, carotenoid, and ubiquinol [184,185]. Enzymatic antioxidants catalyze the removal of ROS to protect the cells [186]. Non-enzymatic antioxidants neutralize the oxidant effect by promoting anti-oxidative enzyme activity or directly processing oxidative chain reactions [185]. Vitamin A, for example, can bind directly to peroxyl radicals before lipid peroxidation can occur. Coenzyme Q10 replenishes vitamin E and counteracts the oxidative effects of lipid peroxyl radicals [187]. The superoxide radical anion, hydrogen peroxide, hydroxyl radical, singlet oxygen and reactive nitrogen oxide can all be effectively neutralized by vitamin C (ascorbate) [188]. These antioxidants prevent further damage by donating free electrons to the cells, thereby preventing oxidation of oxidizable substrates [182]. Through the transsulfuration pathway, in the liver (and other organs), there is endogenous production of GSH, a potent antioxidant that shields cells from oxidative damage [189] and provides immune system support, DNA synthesis and repair and detoxification, thus restoring other critical cellular functions [190,191,192]. Especially critical are mitochondrial GSH (mGSH) levels that serve an important role in maintaining an appropriate mitochondrial redox environment, thereby avoiding/repairing oxidative modifications that lead to mitochondrial dysfunction and cell death [191]. Studies have reported that mGSH emerges as a primary defense against oxidative damage to mitochondrial membranes by ensuring the reduction of hydroperoxide groups on phospholipids and other lipid peroxides through the actions of mitochondrial GSH-S-transferases (GSTs) [193,194]. However, there is a selective depletion of mGSH after alcoholic liver injury, as reported by Checa et al. [195,196]. In the context of liver injury, understanding how decreased antioxidants result in increased ROS is important because oxidant stress contributes significantly to the development and progression of alcohol-induced liver injury. Chronic liver inflammation and damage impair the organ’s ability to produce and utilize antioxidants, such as GSH, leaving liver cells more vulnerable to oxidative damage [197].

### 2.11. Potential Mitochondrial Mechanisms of More Conventional Liver Disease Treatments

The use of chemical agents derived from natural sources as an alternative therapy for liver diseases has long been recognized [198]. Recently, substantial research identifying novel compounds that target mitochondria has focused on the development of drugs that alleviate mitochondrial dysfunction [34]. Augmented production of ROS, owing to mitochondrial dysfunction is associated with progression of NAFLD [199]. Supplementation with avocado oil, which has abundant antioxidants, reportedly ameliorates high-fat diet-induced impairments of complex III activity and regulates the transfer of electrons, thereby reducing oxidant stress and mitochondrial dysfunction [200]. The inclusion of avocado oil in the diets of diabetic rats significantly reduced ETC malfunction and attenuated mitochondrial oxidant stress in their livers [201]. Others report that silybin, isolated from milk thistle seeds attenuates fatty-acid-induced cellular damage by increasing mitochondrial size, improving mitochondrial cristae organization, stimulating mitochondrial respiration, ATP production and fatty acid β-oxidation and lowering collagen content in an *in vitro* model of MASLD/MASH [202,203]. Puerarin, a bioactive isoflavone isolated from the *Pueraria lobata* roots enhanced liver mitochondrial homeostasis in a MASLD mouse model by improving mitochondrial complex I and complex II activities and regulating mitochondrial NAD+ content by activating PARP-1/PI3K/AKT signaling [204]. Similarly, diosgenin was reported to augment β-oxidation of fatty acids and improve mitochondrial function and mitochondrial membrane potential while decreasing ROS and lipid synthesis by up-regulating AMPK/ACC/CPT1A and down-regulating SREBP-1c/FAS signaling in human hepatocytes [205]. Glycyrrhizic acid, extracted from the roots of licorice, attenuated CCl_4_-induced apoptosis through p53-dependent mitochondrial signaling and prevented the development of hepatic fibrosis in rats [206]. 18β-glycyrrhetinic acid, an active metabolite of glycyrrhizin, diminished glycochenodeoxycholic-acid induced necrosis and apoptosis by down-regulating apoptotic markers, including caspase 10, caspase 3 and PARP, and by inactivating JNK [207]. Moreover, both glycyrrhizin and 18β-glycyrrhetinic acid were reported to prevent mitochondrial permeability transition, the production of ROS and cytochrome c release [207]. Recently, Jin et al. found that glycyrrhetinic acid, which has a mitochondrial-targeting function, inhibits serine hydroxymethyltransferase 2 (SHMT2), a mitochondrial enzyme, thereby downregulating mitochondrial OXPHOS and fatty acid β-oxidation and inhibiting cancer cell proliferation and growth [208].

Clinical and animal studies have shown that betaine (trimethyl glycine), a metabolite of the methionine cycle, and a methylating agent, prevents/attenuates liver injury of various etiologies [209,210,211,212,213,214,215,216]. Dietary supplementation with betaine was found to attenuate alcohol-induced steatosis and steatohepatitis [215,216]. Treatment of laboratory rodents or ethanol-metabolizing HepG2 cells with betaine had a protective effect against ethanol-induced oxidant stress and the disruption of mitochondrial OXPHOS systems [215,217]. Also, betaine protected the liver by regulating mitochondrial dynamics and attenuating oxidant stress during thioacetamide-induced hepatic injury [214]. Moreover, betaine was reported to have an important role in remodeling mitochondrial function by modulating mitochondrial fusion and fission proteins, including dynamin-related protein 1 (DRP1) and mitofusins (MFN2s). Betaine treatment also enhanced cell survival in oligomycin- and rotenone-treated human liver cells [218].

## 3. Conclusions

Chronic liver diseases are among the major global health problems. The pathogenesis of MASLD and ALD has various causes, including genetic, environmental and nutritional factors. Currently, the treatments for liver diseases are preventive and most effective at the early stages of disease. Thus, there is a need for a better understanding of the pathogenic mechanisms, which is essential for developing effective therapies. Here, we implicate mitochondrial dysfunction as a primary pathogenic mechanism in the development of chronic liver diseases, including MASLD and ALD. In addition to causing fat accumulation, mitochondrial dysfunction may also result in an increased generation of cytokines and ROS, which potentiate MASLD and ALD progression by worsening inflammation and fibrosis in the liver. Hence, targeting mitochondrial dysfunction offers a unique therapeutic intervention for treating chronic liver diseases.

## Figures and Tables

**Figure 1 biology-12-01311-f001:**
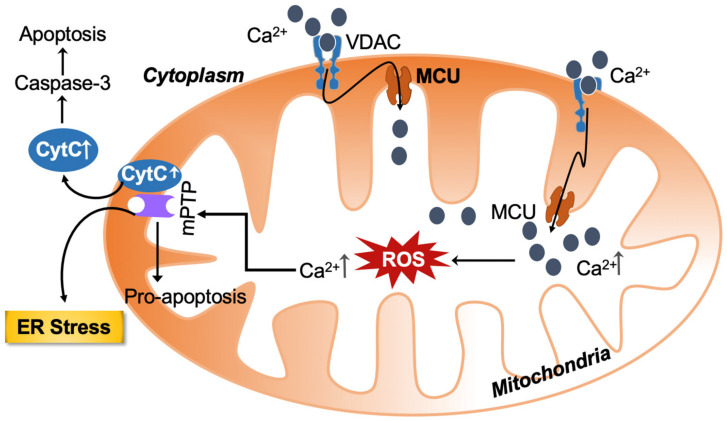
Calcium and ROS-mediated crosstalk between mitochondria and the endoplasmic reticulum (ER) amplify the deleterious cellular effect. The increased ROS and calcium load-mediated opening of the mitochondrial permeability transition pore causes the release of pro-apoptotic substances. ROS can also target ER-based calcium channels, increasing calcium release and ROS levels in the process. Calcium—Ca^2+^; VDAC—voltage-dependent anion channel; MCU—mitochondrial calcium uniporter; ROS—reactive oxygen species; mPTP—mitochondrial permeability transition pore; CytC—cytochrome c; ER—endoplasmic reticulum.

**Figure 2 biology-12-01311-f002:**
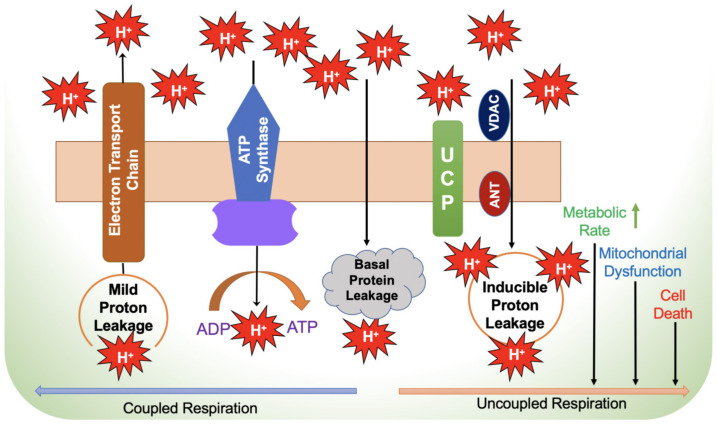
Mechanism of electron leakage during coupled and uncoupled respiration. ADP—adenosine diphosphate; ANT—adenine nucleotide translocator; ATP—adenosine triphosphate; UCP—uncoupling protein; VDAC—voltage-dependent anion channel.

**Figure 3 biology-12-01311-f003:**
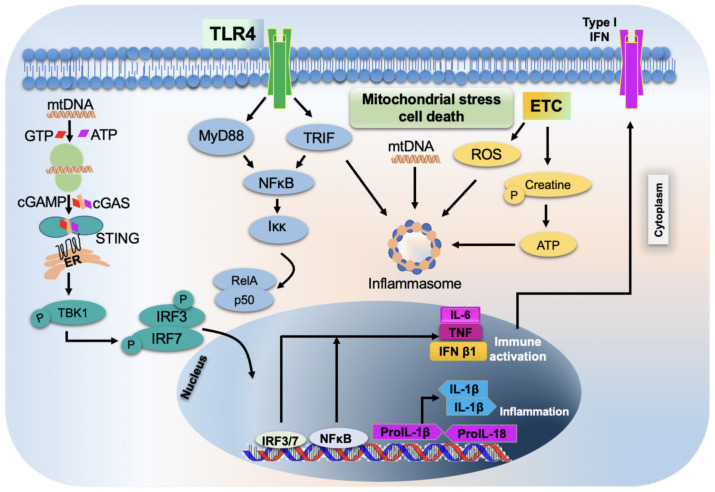
Schematic representation of mitochondrial stress-induced inflammatory response and cell death. TLR4s—toll-like receptors 4; ETC—electron transport chain; IFNs—interferons; IL-6—interleukin 6; TNF—tumor necrosis factor; IFN β1—interferon beta 1; IL-1β—interleukin 1 beta; IRF—interferon regulatory factor; NFκB—nuclear factor kappa beta.

**Figure 4 biology-12-01311-f004:**
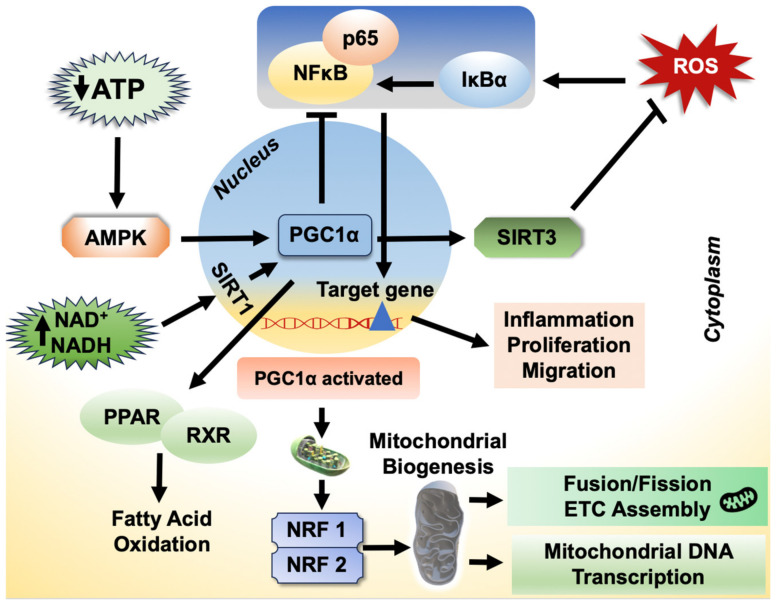
Schematic representation of the role of PGC-1α and its regulatory mechanisms in the involvement of liver diseases. AMPK—AMP-activated protein kinase; SIRT1—sirtuin 1; PPAR—peroxisome proliferator-activated receptors; RXR—retinoid X receptor; NRF—nuclear respiratory factor 2.

**Table 1 biology-12-01311-t001:** Characterization and consequences of mitochondrial dysfunction contributing to the development of chronic liver diseases.

Condition	Characterization	Regulation	Abnormalities	References
Hepatic iron overload	Influences the conversion of hydrogen peroxide (H_2_O_2_) to highly toxic hydroxyl radicals (HO•).	Hepcidin is feedback-regulated by iron concentrations in plasma and the liver and by erythropoietic demand for iron.	ROS accumulation, cytotoxicity and oxidative stress.	[25]
Ferritin overload	Primary intracellular iron storage protein in both prokaryotes and eukaryotes.	Participates in oxidation–reduction, iron ion transport across membranes and cellular iron ion homeostasis.	Hemochromatosis or hemosiderosis.	[26,27]
Chronic iron overload	Enhances iNOS synthase.	Activation of extracellular signal-regulated kinase (ERK1/2) and nuclear transcription factor (NFκB) in the liver.	Liver steatosis and fibrosis.	[28,29]
Ferroptosis	Intracellular iron-dependent form of cell death that is distinct from necrosis and autophagy.	Accumulation of lipid peroxides.	Neurological dysfunction and cell death.	[36]
Voltage-dependent anion channel 1 (VDAC) in outer mitochondrial membrane	Cellular Ca^2+^ homeostasis by mediating the transport of Ca^2+^ in and out of mitochondria. VDAC1 is highly Ca^2+^-permeable and modulates Ca^2+^ access to the mitochondrial intermembrane space.	Mitochondria-mediated apoptosis by the release of apoptotic proteins.	Increase in calcium into the mitochondria leads to apoptosis.	[5,6]
VDAC oligomerization	VDAC oligomerization inducing mitochondrial outer membrane permeabilization causing mtDNA release.	Mitochondrial stress releases mtDNA into the cytosol, thereby triggering the type Ι interferon (IFN) response.	Regulates Ca^2+^ influx, metabolism, inflammasome activation and cell death.	[43]
Mitochondrial calcium uniporter	Transmembrane protein that allows for the passage of calcium ions from cytosol into mitochondria.	Regulated by MICU1 and MICU2 and plays a fundamental role in the shaping of global calcium signaling and in the control of aerobic metabolism, as well as apoptosis.	Oxidative stress-elevated mitochondrial calcium and its function in neurodegenerative disorders. Hepatic lipid accumulation through MCU/PP4/AMPK molecular signaling mechanism.	[8]
Depolarization of the inner mitochondrial membrane mediated	Allows for antioxidant molecules to exit mitochondria, reducing the organelles’ ability to neutralize ROS.	Caspase-mediated apoptosis.	Increased ROS production.	[13]
Paraptosis, a non-apoptotic type of programmed cell death	Non-apoptotic type of programmed cell death.	Fragmentation of DNA and caspase activation, cell death occurring by cytoplasmic vacuolation.	Mitochondrial swelling.	[19]

## Data Availability

Not applicable.

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
