# Peer review of "Mitochondrial Dysfunction-Associated Mechanisms in the Development of Chronic Liver Diseases"

_biology, 2023, doi:10.3390/biology12101311_

Round 1

Reviewer 1 Report

Arumugam et al.

In this manuscript Arumugam et al. review how mitochondrial dysfunction promotes chronic liver disease. The review is mostly comprehensive and well-written, but there is one topic that could use more explanation.

Inflammation plays a very important role in development of chronic liver disease. The Type I interferon pathway and NLRP3 inflammasome pathway (Xian Immunity 2022). In both cases, MOMP causing mtDNA released is created by VDAC oligomerization (first reported in Kim et al. Science 2019). This aspect could be also mentioned in Table 1, in the description of VDAC.

Minor correction: Fig. 4.

ATP decrease, not increase, activates AMPK (which was misspelled).

English quality is adequate.

Author Response

Reviewer 1

In this manuscript Arumugam et al. review how mitochondrial dysfunction promotes chronic liver disease. The review is mostly comprehensive and well-written, but there is one topic that could use more explanation.

Inflammation plays a very important role in development of chronic liver disease. The Type I interferon pathway and NLRP3 inflammasome pathway (Xian Immunity 2022). In both cases, MOMP causing mtDNA released is created by VDAC oligomerization (first reported in Kim et al. Science 2019). This aspect could also be mentioned in Table 1, in the description of VDAC.

Our Response: We have mentioned this aspect in Table 1 of the revised manuscript.

Minor correction: Fig. 4.

ATP decreases not increases, activates AMPK (which was misspelled).

Our Response:   Figure 4 has been replaced by a new figure (Fig. 4) in our revised submission.

Reviewer 2 Report

Madan Kumar Arumugam et al. reviewed the role mitochondrial dysfunction plays in the development and progression of chronic liver diseases, namely, alcoholic (ALD) and non-alcoholic fatty liver disease (NAFLD). The review contains chapters on calcium homoeostasis and overload, hepatic iron overload, increased reactive oxygen species (ROS) production and impaired ROS clearance, the role of mitochondria and mitochondrial dysfunction for inflammation as well as mitochondrial-targeted treatments for inflammatory liver disease. The authors conclude that targeting mitochondrial dysfunction offers a unique therapeutic possibility for the treatment of both ALD and NAFLD.

Comments:

1.        A brief chapter on mitochondrial/cellular calcium homeostasis under physiological conditions should be added before reviewing its dysregulation;

2.        A brief chapter on the efficacy and/or insufficiency of antioxidant treatment for increased ROS production and impaired ROS clearance should be added;

3.        The chapter on mitochondrial-targeted treatments should be expanded a bit to include some information on the potential mitochondrial mechanisms of more conventional liver disease treatments, i.e. glycyrrhizinic acid etc.

English language and style are minor spell check required

Author Response

Reviewer 2

Madan Kumar Arumugam et al. reviewed the role mitochondrial dysfunction plays in the development and progression of chronic liver diseases, namely, alcoholic (ALD) and non-alcoholic fatty liver disease (NAFLD). The review contains chapters on calcium homoeostasis and overload, hepatic iron overload, increased reactive oxygen species (ROS) production and impaired ROS clearance, the role of mitochondria and mitochondrial dysfunction for inflammation as well as mitochondrial-targeted treatments for inflammatory liver disease. The authors conclude that targeting mitochondrial dysfunction offers a unique therapeutic possibility for the treatment of both ALD and NAFLD.

Comments:

  1. A brief chapter on mitochondrial/cellular calcium homeostasis under physiological conditions should be added before reviewing its dysregulation.

Our Response : In the revised manuscript, at the beginning of Section 2.1, we have included a brief paragraph on mitochondrial/cellular calcium homeostasis under physiological conditions. 

A brief chapter on the efficacy and/or insufficiency of antioxidant treatment for increased ROS production and impaired ROS clearance should be added.

Our Response : We thank the reviewer for this suggestion. We added Section 2.10 on “the insufficiency of antioxidants and impaired ROS clearance” into the revised manuscript.    

  1. The chapter on mitochondrial-targeted treatments should be expanded a bit to include some information on the potential mitochondrial mechanisms of more conventional liver disease treatments, i.e., glycyrrhizinic acid etc.

Our Response : Thank you for the valuable suggestion. In the revised manuscript. We added a separate section 2.11 on “Potential Mitochondria-targeted Treatments for Common Liver Diseases”

Reviewer 3 Report

This manuscript has discussed a large volume of literature related to ROS induced mitochondrial dysfunction in the setting of NAFLD and alcohol-associated liver disease. The included table and figures generally summarise the key points well. However, I have some comments that I would like addressed prior to publication:

1.       Both the title and abstract contain no mention of ROS and thus I would suggest that the authors be more specific in both to better reflect the focus of the manuscript.

2.       Section 2 appears to be missing a heading. ‘Calcium overload increases ROS production’ really should be a subheading as it only reflects the subsequent three paragraphs. In addition, these paragraphs appear mostly to discuss the mitochondria’s role in cell death, which again isn’t in the subheading.

3.       Figure 1, the revised figure 1 I received suggests that increased ROS release causes ER stress. But the text describes a situation where ER stress results in increased Ca2+ that is taken up by the mitochondria, which then increases ROS, which then causes apoptosis. In the text there is no mention of an additional effect of this increase in ROS also leading to ER stress. Please adjust the figure and/or text for clarity and completeness.

4.       Section 2.1, perhaps adjust the subheading to better reflect the content of this section.

5.       Section 2.2, is mitochondrial function really impaired or just not sufficient to deal with the increased nutrient load?

6.       Section 2.5, the first sentence should be altered to reflect the new information.

7.       Reference 71 (line 271 page 7) doesn’t appear to be appropriate in this context.

Author Response

Reviewer 3

This manuscript has discussed a large volume of literature related to ROS induced mitochondrial dysfunction in the setting of NAFLD and alcohol-associated liver disease. The included table and figures generally summarise the key points well. However, I have some comments that I would like addressed prior to publication:

  1. Both the title and abstract contain no mention of ROS and thus I would suggest that the authors be more specific in both to better reflect the focus of the manuscript.

Our Response  : In the revised abstract, we have included information on ROS-mediated mitochondrial dysfunction.

  1. Section 2 appears to be missing a heading. ‘Calcium overload increases ROS production’ really should be a subheading as it only reflects the subsequent three paragraphs. In addition, these paragraphs appear mostly to discuss the mitochondria’s role in cell death, which again isn’t in the subheading.

Our Response  :   We have modified the heading of Section 2.1 in the revised manuscript.

  1. Figure 1, the revised figure 1 I received suggests that increased ROS release causes ER stress. But the text describes a situation where ER stress results in increased Ca2+ that is taken up by the mitochondria, which then increases ROS, which then causes apoptosis. In the text there is no mention of an additional effect of this increase in ROS also leading to ER stress. Please adjust the figure and/or text for clarity and completeness.

Our Response  : Thanks for your comments.  We adjusted  figure 1 to match the text in which the increased ER Stress results in increased Ca2+ and ROS, which, in turn, leads to more ER Stress to perpetuate the cycle. The changes were incorporated into Figure 1 in the revised manuscript.

  1. Section 2.1, perhaps adjust the subheading to better reflect the content of this section.

Our Response  : We edited Section 2.1 subheading, in accordance with the reviewers’ suggestions.

  1. Section 2.2, is mitochondrial function really impaired or just not sufficient to deal with the increased nutrient load?

Our Response  :  We edited the section 2.2, and replaced it with a new section (Section 2.3)  on “ Excessive carbohydrates and fatty acids intake”. There are many factors involved in the impairment of mitochondrial biogenesis. Among those, intake of excess carbohydrates and fatty acids are two of the major causative factors, which  reportedly increase de novo lipogenesis in the liver. In connection with this, iron and calcium overdose cause mitochondrial dysfunction, leading to accumulation of lipids which increases oxidant stress and steatosis in hepatocytes.  The latter  further proceeds to  steatohepatitis.

  1. Section 2.5, the first sentence should be altered to reflect the new information.

Our Response  : We thank you for that suggestion. We have altered the first sentence in   section 2.5 and it is reframed in the revised manuscript.

  1. Reference 71 (line 271 page 7) doesn’t appear to be appropriate in this context.

Our Response  : We made appropriate corrections in the revised manuscript in accordance with each reviewer’s suggestion.

Round 2

Reviewer 2 Report

The authors took into account comments and suggestions.